# Hyperparameter Optimization of Bayesian Neural Network Using Bayesian Optimization and Intelligent Feature Engineering for Load Forecasting

**DOI:** 10.3390/s22124446

**Published:** 2022-06-12

**Authors:** M. Zulfiqar, Kelum A. A. Gamage, M. Kamran, M. B. Rasheed

**Affiliations:** 1Department of Telecommunication Systems, Bahauddin Zakariya University, Multan 60000, Pakistan; zulfiqarchishti@gmail.com; 2James Watt School of Engineering, James Watt South Building, University of Glasgow, Glasgow G12 8QQ, UK; 3Department of Electrical Engineering, University of Engineering and Technology, Lahore 54000, Pakistan; mkamran@uet.edu.pk; 4Escuela Politécnica Superior, Universidad de Alcalá, ISG, 28805 Alcalá de Henares, Spain; muhammad.rasheed@uah.es

**Keywords:** Bayesian Neural Networks, Bayesian Optimization, convergence rate, Hamilton dynamic, electric load forecasting

## Abstract

This paper proposes a new hybrid framework for short-term load forecasting (STLF) by combining the Feature Engineering (FE) and Bayesian Optimization (BO) algorithms with a Bayesian Neural Network (BNN). The FE module comprises feature selection and extraction phases. Firstly, by merging the Random Forest (RaF) and Relief-F (ReF) algorithms, we developed a hybrid feature selector based on grey correlation analysis (GCA) to eliminate feature redundancy. Secondly, a radial basis Kernel function and principal component analysis (KPCA) are integrated into the feature-extraction module for dimensional reduction. Thirdly, the Bayesian Optimization (BO) algorithm is used to fine-tune the control parameters of a BNN and provides more accurate results by avoiding the optimal local trapping. The proposed FE-BNN-BO framework works in such a way to ensure stability, convergence, and accuracy. The proposed FE-BNN-BO model is tested on the hourly load data obtained from the PJM, USA, electricity market. In addition, the simulation results are also compared with other benchmark models such as Bi-Level, long short-term memory (LSTM), an accurate and fast convergence-based ANN (ANN-AFC), and a mutual-information-based ANN (ANN-MI). The results show that the proposed model has significantly improved the accuracy with a fast convergence rate and reduced the mean absolute percent error (MAPE).

## 1. Introduction

An accurate ELF is essential for smart grids (SGs) to make strategic decisions such as operational and planning management [1], load switching [2], energy generation expansion, maintenance scheduling, security, demand monitoring inspections, and providing a reliable energy supply [3], since inaccurate forecast results may pose serious challenges in making short- and long-term decisions and planning for SGs. Overestimation in a forecast may lead to excessive spinning reserves, production capacity, and limited energy distribution, resulting in higher operational costs. In contrast, underestimation may create consistency, power quality, safety, and monitoring issues. Therefore, the distribution system operators (DSOs) need an acceptable accuracy to guarantee endurance and stable grid operation [4]. For this purpose, much devotion is given to provide instant, accurate, and stable load forecasts to ensure the safe and reliable operation of the grid [5]. However, the accuracy of ELF often cannot cope with societal requirements. It is affected by probabilistic and uncertain factors such as economic development, human social activity, irradiance meteorological parameters, environmental parameters, temperature, climate change, and government policies [6]. Therefore, improving the accuracy of the predictive network is a challenge. Considering all influencing factors is unrealistic or cumbersome [7]. Consequently, we can improve prediction accuracy by developing a model that considers the key parameters. On the other hand, different forecasting algorithms have been developed to accurately predict the load. These include the time-series models such as Kalman filter-based approaches [8], the box–Jenkins model [9], state-space models [10], exponential soothing [11], regression-based methods [12], and the autoregressive integrated moving average (ARIMA) [13]. Traditional statistical strategies identify and capture performance patterns and apply a time-series approach for better results. However, these techniques are useful for linear analysis and are unable to deal with nonlinear time-series problems, which is where artificial intelligence (AI) methods have also been commonly used to improve performance [14]. As these models are not involved in complex mathematical modeling, therefore, they can easily and properly process the historical load-demand data, effectively and efficiently [15]. The most commonly used AI-based approaches include expert systems [16], support vector machines (SVMs) [17], machine learning (ML) models [18], deep learning (DL) models [19], and artificial neural networks (ANNs) [20] that can be used for ELF. Among other forecasting techniques, an ANN is one of the most widely used algorithms due to its improved accuracy. However, an ANN has problems with network design, overfitting, history dependencies, and connection-weight estimation. However, in contrast, a BNN has recently received attention due to its excellent performance and capability to handle uncertain tetrameters [21]. This is due to its learning capability from the network architecture and the selection of weight vector, while network topology is comprised of three feed-forward layers, with an arbitrary number of hidden units. Then, by taking into account the network design, learning the weight vector entails the selecting of a weight vector with the largest posterior probability distribution [22,23,24]. Load forecasting time series typically consists of prefiltering, sequence modeling, independent forecasting, and aggregation. Therefore, a large number of hyperparameters need tuning for higher forecast accuracy. A grid search is mostly considered to change the configurations, due to its simplicity. However, repetitive searches without correlation can be extraordinarily time-consuming and biased. When faced with the tuning task, evolutionary and genetic algorithms become the choices for interpretability. The motivation for considering the BO algorithm [25] in hyperparameter tuning lies with two aspects. Firstly, the time cost is controllable and can be fixed according to our necessities. Secondly, the balance of exploitation and exploration is well-sustained compared to other existing models, making the search more effective. Ref. [26] implements the BO algorithm for the attention-based LSTM model for stream-flow prediction. Ref. [27] optimizes the network structure and the time lag for prediction. Similarly, [28] builds a hybrid model combining multi-ahead attention, LSTM, and CNN outputs, applying the BO algorithm to search for optimal model hyperparameters for COVID-19 prediction. All the works mentioned above validate the improvement of using the BO algorithm compared with baseline models.

The combined model of predictions is still inconsistent due to some open issues. The first drawback occurs when the model candidates change a little. For example, [29] only evaluates candidates’ deep neural networks (DNN) such as multilayer perceptron (MLP), a convolutional neural network (CNN), and long short-term memory (LSTM). Ref. [30] combines only MLP, dynamic architecture for an ANN, a recurrent neural network (RNN), and an echo state network (ESN). Authors in [31] conclude that pure data-driven ML and DNN models are not well-suited for large-scale predictive competition. Combining the statistical and DNN predictive models provides more robust and effective predictions. An adaptive combinational method that assigns time-varying weights to the model to solve the underlying pattern shifts provides promising forecasts. However, some vital signs are delicate and prevent practitioners from using them efficiently. They must be adjusted manually. Last but not least, the performance of many combinational strategies has not been fully confirmed due to the small amount of experimental data from related studies and the seasonal diversity.

Hence, a new feature engineering (FE) and optimization concept is introduced. The proposed BO algorithm has been selected to adjust the control parameters because of its fast convergence and robustness in finding the optimal solution [32,33,34]. The BO algorithm optimizes the threshold weights for the filters and finds the optimized thresholds to be used in the FE module for feature selection. The complete forecasting framework consists of an integrated framework of feature engineering (FE), a stochastic BNN model, and the BO algorithm (FE-BNN-BO). The performance of the proposed FE-BNN-BO model is validated by comparing the results with the existing models in terms of mean absolute percentage error (MAPE).

### 1.1. Contributions

The real contributions are presented as follows:An ingenious and robust framework, FE-BNN-BO has been proposed that integrates the FE and BO algorithm with BNN. The FE module solves the concern associated with redundancy and irrelevance (dimension reduction). In the meantime, the BO algorithm optimizes the hyper-parameters of the BNN predictor to enhance accuracy while securing fast convergence. The combination of the FE module and the BO algorithm significantly improves the performance and effectiveness of the BNN model.BNN models are complex in estimating computational efficiency and cannot handle uncertainty. Therefore, the iterative and irrelevant features may enhance the complexity, slow down the BNN training process and affect the prediction accuracy. The proposed FE addresses this problem by combining the Random Forest and Relief-F-based feature selector and radial-based kernel principal component analysis (RBKPCA)-based feature extractor. The feature selector converges the Random Forest with Relief-F, calculates the importance of the feature, selects the relevant feature, and discards the irrelevant feature. This further enhances the computational performance and efficacy of the BNN model.Moreover, the BO algorithm is automatically applied to search for the best ensemble configuration. The devised BO algorithm is more controllable and efficient in time and complexity than the widely used grid search methods.The proposed model is validated against the latest hourly load data obtained from the USA electricity grid. The proposed frameworks outperformed the benchmark frameworks, such as LSTM, ANN-MI, ANN-AFC, and Bi-Level, when considering the accuracy and convergence speed.

### 1.2. Paper Organization

The rest of the sections of this work are organized as follows: the recent and relevant work is demonstrated in Section 2. Section 3 illustrates the proposed system model. Section 4 discusses the simulation results and discussion. The paper is finalized in Section 5.

## 2. Literature Survey

This is because static and ML models are widely used in the literature. These can be divided into two major categories to understand better how they perform and the impediments that come with them. A detailed description is as follows.

### 2.1. Individual ELF Models

The individual models are used for ELF without fusing any other algorithm. Therefore, only the algorithmic efficacy is estimated using various performance parameters. The authors [35] have proposed a distribution practice for meteorological data predicting the prospective load. The energy system is further divided into two sub-systems depending on the climate. In addition, the two distinct forecasting models, Grey and ARIMA, are used in both sub-systems. The fitted models are assessed by approximating them to the definitive models using MAPE as a performance metric. In [36], an individual approach based on a deep recurrent neural network (DRNN) is introduced to forecast the household load. This approach ensures the overfitting issue more efficiently than the classical DNN systems. Furthermore, the results show the improved performance of the proposed strategy by comparing it to the other single methods, such as ARIMA, SVM, Grey, and traditional RNNs. Authors in [37] proposed an RNN based on the long short-term memory (RNN-LSTM) framework to forecast the household loads. The forecast accuracy has been enhanced by utilizing the embedded appliance-usage series strategy of training data. However, the author ignores the convergence rate and computational complexity, and they focus only on accuracy. The demand response scheme has considered the ANN for the price forecasting in [38]. The proposed price-forecasting model uses mixed-integer linear programming (MILP) to lessen energy costs. Simulation results depict that hourly demand response is more optimistic than a day ahead, with an enhanced ability to encounter the industrial market by diminishing cost. The authors presented a probabilistic prediction model for predicting PV, electrical energy consumption, and scalability [39]. Quantile regression (QR) models and dynamic Gaussian processes (GP) are applied to the Sydney metropolitan area data for probabilistic prediction. Simulation results demonstrate that the proposed model excels in all three predictive scenarios. In [40], a long-term predictive model was proposed to improve the relative prediction accuracy of the integrated power resource plan. The authors [41] have investigated new aspects using loads and temperatures over the past few hours. The primary purpose is to determine the hourly moving average temperature with a time lag to improve the accuracy of the prediction. The impact of timeliness is examined in three scenarios: the aggregated geographic hierarchy level, the lowest geographic hierarchy level, and each time. However, it improves accuracy at the cost of model complexity. Though individual models are robust and fast converging, they are still inaccurate and do not reach the required level.

The above discussion finalizes that single strategies are not helpful in all facets (rate of convergence, accuracy, stability) due to each technique’s unique flaws, imperfections, defects, and intrinsic limitations [41]. For example, non-linear and seasonal behavior cannot be learned from a linear regression-based model [12]. The gray model is distinctive for the exponential growth trend [11]. Expert systems rely on a solid knowledge base [16], and intelligent methods rely on thresholds, weights, biases, and hyperparameter adjustments [14]. These annoyances affect ELF and result in inconsistent performance. Due to these shortcomings, individual methods cannot achieve all goals (accuracy, rate of convergence, stability) simultaneously. Multiple optimization algorithms, such as meta-heuristic [42], bio-inspired [43], and heuristics [44], are integrated into a single model to devise hybrid models to overcome the problems and limitations of the single methods [33]. The goal is to attain increased precision and overwhelm the flux of final forecasts by optimizing thresholds, hyper-parameter adjustments, random weight initialization, and biases for individual models.

### 2.2. Hybrid ELF Models

The new integrated and hybrid predictive model is an intelligent solution that maximizes the desired characteristics of individual models to ensure superior performance [45,46]. The hybrid model integrates the FE engine and the optimization engine with a combination of prediction algorithms to improve accuracy by optimizing the control parameters of the prediction engine. For instance, a hybrid wavelet neural network (WNN) based on a stepped forward differential evolution (SFDE) framework is devised. The optimizing algorithm SFDE efficiently tunes the hyperparameters of the proposed WNN. Experimental results show that the proposed framework is efficient, when compared with different frameworks such as ANN-based particle swarm optimization (ANN-PSO), ANN-based genetic algorithm (GA-ANN), and ANN-based evolutionary programming (ANN-EP) in terms of accuracy, efficiency, effectiveness, and hyperparameters tuning for ELF [47]. A hybrid model of nonlinear AR with GA and an extrinsic NN is proposed in [48] for STLF. Use statistical and pattern-recognition-based schemes to fine-tune the input parameters of the proposed model. GA is used for the weight and bias of the NN training selection. The proposed model is validated by comparing it to existing models such as means and regression tree models with extrinsic inputs. The author proposed a robust STLF framework with an automated data cleaning method for load prediction of distribution feeders [49]. The previous day’s building level LF model was proposed based on DL [50]. The proposed DL model is validated by comparing the accuracy with the traditional models. An integrated framework for VMD, LSTM, and BO algorithms has been proposed [51]. This model aims to be superior to existing models regarding accuracy and stability. A modified hybrid model of the multipurpose cuckoo search algorithm (MCSA) and GNRR has been proposed [52]. The proposed model will be tested against existing models for predictive accuracy using real-time load data from the Australian energy market operator (AEMO). The author [53] proposed a prediction engine based on the Neural Elman network to predict the future load of SG. Intelligent algorithms optimally adjust the biases and weights of this network to acquire accurate predictions. The author [54] proposed an STLF model based on SSVR. The main goal is to enhance the accuracy and efficiency of comparative predictions. The output of the prediction engine passes through the optimization engine and fine-tuning the parameters to improve the accuracy and efficiency of exact predictions. However, the prediction accuracy is enhanced at the expense of computational complexity. A fused framework is presented in [55] based on SVR and DE to enhance the forecasting performance by adapting SVR parameters. The developed framework surpasses backpropagation ANN, regression frameworks, and typical SVR. While in [56], a hybrid of SVR and the fruit-fly (Ff) algorithm framework is designed to address hyperparameter selection and improve forecasting accuracy. In addition, a new approach has been developed to achieve accurate ELF by merging the firefly optimization algorithm (FFO) with the SVR model and fitting optimal hyperparameters in [57,58]. A hybrid prediction strategy has been proposed to solve the above problems. However, the hybrid prediction strategy has improved modeling capabilities compared to the non-hybrid methods. Still, slow convergence and long execution times have a problem due to the many adjustable parameters. In [59], the author used a Bi-Level strategy based on ANN and the DE algorithm for ELF. Methods based on AFC-ANN and the modified extended DE algorithm (MEDEA) [60] have been proposed to predict future loads [61].

The combination of static optimal predictions calculates the weights of each model through pairwise performance fitting of training verifications that have been studied empirically and theoretically unpredictably. The authors have developed a hybrid model consisting of the navel switching delayed PSO (NSDPSO) algorithm and ELM for STLF [62]. Weights and biases are optimized using the proposed NSDPSO algorithm. The tanh function is considered an activation function because it has more generalization issues and avoids unwanted hidden nodes and overfitting issues. Experimental results show that the proposed framework is superior to RBFNN. The devised model successfully applies to the STLF of the energy system. A hybrid prediction framework has been developed that combines a feature extraction method with a two-step prediction engine. The two-stage prediction engine uses Ridgelet NN (RNN) and Elman NN (ENN) to provide accurate predictions. Optimization algorithms are applied to determine the control parameters of the prediction engine [63]. The hybrid models above can be considered optimistic and valuable in improving prediction accuracy by adequately modifying the hyperparameters. However, the authors of these articles focus on bias initialization and random weighting optimization or proper adjustment and selection of hyperparameters. Moreover, none of these models considered accuracy, rate of convergence, and stability simultaneously. From numerous analyses and investigations, one aspect (bias initialization and random weight optimization, or proper setting and selection of hyperparameters) and one measurement (convergence, accuracy, or stability) are not enough. We have concluded that it is sufficient. Therefore, a robust hybrid model is needed to overcome the problems of current models while improving prediction accuracy and stability with fast convergence rates.

From the literature, we can safely conclude that ELF has made great strides in energy management. However, existing approaches are not practical when dealing with large amounts of data. Adjusting control parameters is complex, and redundancy, irrelevance, and dimensionality reduction are unavoidable, which makes the calculation very difficult as it cannot quickly converge. Furthermore, the above literature does not consider forecast accuracy and rate of convergence at the same time. To address these problems, we require a fast and accurate model. Therefore, an SVM and gradient descent (GD) algorithm-based model is proposed [64]. However, this model introduces computational complexity and fails to converge. Some authors have focused on feature selection algorithms, traditional classifier decision trees (DTs), and ANNs [65]. However, the DT faces the problem of overfitting. So, while DT works well in training, it does not work well in prediction. ANNs have limited generalizability and are challenging to control convergence. The authors [41] proposed a hybrid feature selection (HFS), extraction, and classification model for STLF. However, this method is too complex to converge.

## 3. Proposed Model

This study proposes a novel hybrid framework based on the FE method, neural network model (BNN), and BO algorithm for ELF, as shown in Figure 1. This work targets only daily load forecasting using a new concept of scalability and robustness evaluation. The proposed model is an integrated framework of three modules: (i) an FE module comprised of hybrid feature selector (HFS) and the feature extractor (FX), (ii) a forecasting module based on the BNN model, and (iii) an optimizing module based on the BO algorithm.

### 3.1. FE Module

The first module is FE. In this phase, abstract and critical features are picked and removed from the preprocessed data, while repetitious and irrelevant elements are discarded. The desired features are picked from the dataset by GCA and extracted by radial basis KPCA (RB-KPCA). FS relies on GCA to drive feature selection. It includes Relief-F (ReF) and Random Forest (RaF) algorithms for estimating the importance of features as depicted in Figure 2. In addition, the FS decides whether to reserve or abandon a feature based on the extent. RBKPCA-based FX uses kernel functions to process high-dimensional nonlinear data. Feature extraction aims to reduce redundant features. Below is a brief demonstration of the FE module.

#### 3.1.1. FS

The feature selection system is primarily based totally on GCA, which has evolved with the aid of using RaF and ReF and is managed using a combined controlling threshold (φ). The GCA more or less selects a feature space in which the maximum applicable and preferred features are kept, and inappropriate features are discarded primarily based totally on the feature significance and feature selection φ. Let L be the electric load data matrix, which is defined as follows: (1)$$L=\left[\begin{array}{ccccc} l_{11} & l_{12} & l_{13} & \cdots & l_{1n}\\ l_{21} & l_{22} & l_{23} & \cdots & l_{23}\\ \vdots & \vdots & \vdots & ⋰\ddots & \vdots \\ l_{m1} & l_{m2} & l_{13} & \cdots & l_{mn} \end{array}\right]$$

The columns show the feature index, while the rows present the timestamps. Moreover, $l_{mn}$ is the mth component of data, which is nth hour ahead of the electrical energy consumption pattern that is to be forecasted. Equation (Equation 1) can also be expressed in the form of a time sequence:(2)$$L=\left[\begin{array}{c} t_{1}\\ t_{2}\\ \vdots \\ t_{n} \end{array}\right]\;$$
where,
(3)$$t_{k}=\left[l_{k1}\;\;l_{k2}\;\;l_{k3}\;\cdots \;\;\;\;l_{kn}\right]\;k\in \left[1,m\right].$$

Many factors affect the ELF pattern in different ways. GCA estimates the importance of each component and its impact on ELF. GCA effectively controls the feature selection process by determining the correlation between each feature and the final ELF pattern. GCA resolves the accessibility of data signals by correlation. Correlation is directly correlated to the proximity of the data signal. As a result, GCA determines the proximity of the two data signals. Each framework feature has its physical meaning and dimensions, so dimensionless data is standardized by the mean or maximum when GCA is executed. Since the original sequence has the characteristic of “the larger the better” [66], it can be normalized as:   
(4)$$Z_{j}^{*}\left(k\right)=\frac{Z_{j}\left(k\right)-{\mathrm{Zmin}}_{j}\left(k\right)}{{\mathrm{Zmax}}_{j}\left(k\right)-{\mathrm{Zmin}}_{j}\left(k\right)}$$
where $Z_{j}\left(k\right)$ is the original sequence, $Z_{j}^{*}\left(k\right)$ is the sequence after the data preprocessing, ${\mathrm{Zmax}}_{j}\left(k\right)$ is the largest value of $Z_{j}\left(k\right)$, and ${\mathrm{Zmin}}_{j}\left(k\right)$ is the smallest value of $Z_{j}\left(k\right)$. The grey rational coefficient (η) after normalization [66] is calculated in Equation (Equation 5) as follows:(5)$$\eta \left(Z_{0}^{*}\left(k\right),Z_{j}^{*}\left(k\right)\right)=\frac{\Sigma_{\min}+{\nu \Sigma }_{\max}}{\Sigma_{0j}\left(k\right)+{\nu \Sigma }_{\max}},\;\nu \in \left(0,1\right)$$
where $\Sigma_{0j}\left(k\right)$ is the deviation sequence of the reference sequence $Z_{0}^{*}\left(k\right)$ and the comparability sequence $Z_{j}^{*}\left(k\right)$, and ν is a distinguishing coefficient, fixed to 0.50 [67] and represented in Equation (Equation 6):(6)$$\begin{array}{c} \Sigma_{oi}\left(k\right)=\left.\left|Z_{0}^{*}\left(k\right)-Z_{j}^{*}\left(k\right)\right.\right|,\\ \Sigma_{\max}=\max_{j,k}\left.\left|Z_{0}^{*}\left(k\right)-Z_{j}^{*}\left(k\right)\right.\right|,\\ \Sigma_{\min}=\min_{j,k}\left.\left|Z_{0}^{*}\left(k\right)-Z_{j}^{*}\left(k\right)\right.\right| \end{array}$$

The grey relational grade ($G_{j}$) [68,69] is a weighting sum of the η. It is defined in Equation (Equation 7) as follows:(7)$$G_{j}\left(Z_{0}^{*},Z_{j}^{*}\right)=\frac{\sum_{k=1}^{m}\eta \left(Z_{0}^{*}\left(k\right),Z_{j}^{*}\left(k\right)\right)}{m}$$

Grey relational analysis is a measurement of the absolute value of the data difference between sequences, and it can be used to calculate approximation correlation. The low-correlated features are deleted, and the remainder of the selected items $l_{kn}$ are arranged from least to most significant, providing the time sequence $t_{j}$ as illustrated in Equation (Equation 8):(8)$$t_{j}=\left[l_{k1}\;\;l_{k2}\;\;l_{k3}\;\cdots \;\;\;\;l_{kn-\delta }\right],$$
where δ illustrates the dropped function and $t_{j}$ is the time series.

The RaF evaluator β processes the boot-strap-bagging (BSB) samples [70]. BSB samples are split into out-of-bat (OoB) samples and training samples. In the first evaluator β, all weights are initialized to zero, and RaF training begins. Then, the feature’s importance is determined by the OoB data with noise. For the second evaluator α, the weights are updated with the concept of distance among hits and misses. Both α and β evaluators forward the determined feature importance to the FS to perform feature selection based on the φ.

The $F_{i}^{f}\left[\tau_{k}\right]$ and $F_{i}^{r}\left[\tau_{k}\right]$ represents the feature importance calculated by ReF and RaF, respectively. The parameters are updated in Equations (Equation 9) and (Equation 10), respectively:(9)$$F_{i}^{r}\left[\tau_{k}\right]=F_{i}^{r}\left[\tau_{k}\right]-\frac{\sum_{j=1}^{k}ϝ\left(D,\;b^{*}\right)}{n*k}$$
(10)$$\begin{array}{c} F_{i}^{f}\left[\tau_{k}\right]=F_{i}^{f}\left[\tau_{k}\right]-\frac{\sum_{j=1}^{k}ϝ\left(D,\;b^{*},\;L_{j}\right)}{m*k}\\ +\;\frac{\sum_{C\neq class(x)}ϝ\left(D,\;b^{*},\;N_{j}\left(C\right)\right)}{m*k}, \end{array}$$
where *C* is the class and $b^{*}$ is the randomly selected item in the *C*, and function $ϝ\left(D,\;b^{*},\;N_{j}\left(C\right)\right)$ computes the attribute difference *D* between $r_{1}$ and $r_{2}$. The function ϝ is mathematically modeled as in Equation (Equation 11):(11)$$ϝ=\left\{\begin{array}{c} 0\;values\;are\;distinct\\ 1\;values\;are\;same\\ Difference\;among\;attributes\;is\;\\ normalized\;to\;be\;within\;[01] \end{array}\right.$$

The ReF-based feature importance $F_{i}^{f}$ and RaF-based feature importance $F_{i}^{r}$ are normalized for feature selection, as depicted in Equations (Equation 12) and (Equation 13):(12)$$\bar{F_{i}^{r}}=\frac{F_{i}^{r}}{\max\left(F_{i}^{r}\right)}$$
(13)$$\bar{F_{i}^{f}}=\frac{F_{i}^{f}}{\max\left(F_{i}^{f}\right)}$$

A combined feature importance value greater than φ is considered as a key feature, while those with value less than φ are rendered irrelevant. The core features are restored, while the irrelevant features are eliminated. This procedure is mathematically represented in Equation (Equation 14):(14)$$\tau_{k}=\left\{\begin{array}{c} retain\;F_{i}^{r}\left[\tau_{k}\right]+F_{i}^{f}\left[\tau_{k}\right]>\sigma \\ remove\;F_{i}^{r}\left[\tau_{k}\right]+F_{i}^{f}\left[\tau_{k}\right]\leq \sigma \end{array}\right.$$

The selected features are passed to a feature extraction phase that uses RBKPCA to reduce redundancy between features.

#### 3.1.2. FX

The feature extraction procedure based on RB-KPCA is committed in the second stage. This operation aims to remove redundant data to solve the dimensionality-lessening problem. The output of the FS is sent to the RB-KPCA-based FX. This produces a dimensionally diminished matrix presented in Equation (Equation 15), including the most relevant features of interest that can be modeled as in [71]:(15)$$R={\left(r_{1},r_{2},\;r_{3},\ldots ,r_{j}\right)}^{T}$$
where $r_{j}$ is the jth variable associated with the EL. The correlation between eigenvalues and features is calculated as follows:(16)$$\lambda_{ev}=V^{f^{*}}ev,\;\lambda \geq 0\;\&\;ev\in f^{*}$$
where the covariance matrix of R is denoted by V, and $f^{*}$ represents the feature space, while the eigenvector is represented by ev and λ is the eigenvalue. Furthermore, $V^{f^{*}}ev$ is determined using Equation (Equation 17):(17)$$V^{f^{*}}ev=\frac{1}{M}\sum_{j=1}^{M}\langle \phi \left(r_{j}\right),\;ev\rangle \phi \left(r_{j}\right),$$
while
(18)$$\sum_{k=1}^{M}\phi \left(r_{k}\right)=0,$$
where ϕ shows the feature space and input data mapping, and $\langle r,z\rangle$ expresses the product of *r* and *z*. Equation (Equation 16) becomes Equation (Equation 19) by proposing the above-mentioned modifications:(19)$$\lambda \langle \phi \left(r_{k}\right),ev\rangle =\langle \phi \left(r_{k}\right),V^{f^{*}}ev\rangle ,$$
where ev for λ=0 can be determined as in Equation (Equation 20):(20)$$ev=\sum_{j=1}^{M}\gamma_{j}\phi \left(r_{j}\right),$$
where $\gamma_{j}$ denotes indices corresponding to $r_{j}$. The kernel function mentioned in [72] is now utilized as follows in Equation (Equation 21):(21)$$K_{jk}=\langle \phi \left(r_{j}\right),\;\phi \left(r_{k}\right)\rangle \;\forall \;j,\;k\in \left[1,\;M\right]$$
after combining Equations (Equation 19) and (Equation 20), the combined form is defined as:(22)$$\lambda \sum_{j=1}^{M}\gamma_{j}K_{j}=\frac{1}{M}\sum_{j=1}^{M}\gamma_{j}\sum_{k=1}^{M}K_{kj}K_{jk},$$
where
(23)$$\gamma ={\left[\gamma_{1},\gamma_{2},\ldots ,\gamma_{M}\right]}^{T}$$

Now, Equation (Equation 19) may be rewritten as:(24)$$\lambda MK\gamma =K^{2}\gamma$$

To conduct dimensionality reduction by normalization, the eigenvectors γ and λ are chosen. Therefore, we have:(25)$$\langle ev_{j},\;ev_{k}\rangle =1\;\forall \;j,\;k\in \left[1,\;M\right].$$

The consequential Equation (Equation 26) can be achieved by substituting Equation (Equation 20) for Equation (Equation 25), which is as follows:(26)$$\;\langle \sum_{j=1}^{M}\gamma_{j}^{n}\phi \left(r_{j}\right),\;\sum_{k=1}^{M}\gamma_{k}^{n}\phi \left(r_{k}\right)\rangle =1$$

The LHS of Equation (Equation 26) is further solved in Equations (Equation 27)–(Equation 30) to prove the Equation (Equation 25):(27)$$=\sum_{j=1}^{M}\sum_{k=1}^{M}\gamma_{j}^{n}\gamma_{k}^{n}\langle \phi \left(r_{j}\right),\;\phi \left(r_{k}\right)\rangle$$
(28)$$=\sum_{j=1}^{M}\sum_{k=1}^{M}\gamma_{j}^{n}\gamma_{k}^{n}K_{jk}$$
(29)$$=\langle \gamma_{n},\;K\gamma_{n}\rangle$$
(30)$$=\lambda_{n}\langle \gamma_{n},\;\gamma_{n}\rangle$$

The principal component extraction can be calculated in Equation (Equation 31):(31)$$P_{n}=\langle ev_{n},\;\phi \left(r\right)\rangle =\sum_{j=1}^{M}\gamma_{j}^{n}\langle \phi \left(r_{j}\right),\;\phi \left(r\right)\rangle ,$$
where P signifies the principal element and the generalized versions of the kernel function are:Linear kernel function: the linear kernel is used when the data is linearly separable. It can be separated by one line. This is one of the most commonly used kernels. This is primarily used when a particular dataset contains a large number of features. Mathematically, it may be formulated in Equation (Equation 32):
(32)$$K\left(r,\;z\right)=\langle r,\;z\rangle$$

Kernel function based on logistic sigmoidal: this function is equivalent to a two-layer, perceptron model of the neural network, which is used as an activation function for artificial neurons. Equation (Equation 33) show the mathematical representation of kernel-based sigmoid function:
(33)$$K\left(r,\;z\right)=\tanh\left(a_{0}{\langle r,\;z\rangle }^{d}+a_{1}\right)$$

Kernel function based on radial basis: radial basis function kernels or RBF kernels are common kernel functions used in various kernel-learning algorithms. In particular, they are often used to classify SVMs. Mathematically, an RBF kernel is represented in Equation (Equation 34):
(34)$$K\left(r,\;z\right)=\exp\left(-\theta {∥r-z∥}^{2}\right)$$

After the FE step, the selected and extracted feature matrix is provided as input to the BNN-based prediction engine for predicting performance patterns.

### 3.2. BNN-Based Forecasting Module

The fundamental purpose of NN training is to obtain an appropriate network architecture A and weight vector w. NN offers an implicit function f(x,w) design that connects the input variable x to the output variable y provided as A and w. For the dataset $D=\left(x_{1},\tau_{1}\right),\left(x_{2},\tau_{2}\right),...\left(x_{n},\tau_{n}\right)$, with assumed A, to achieve w with reference to the weight vector by training it, the mapping function f(x,w) has the lowest error $E_{D}\left(w\right)$ [73] and is presented in Equation (Equation 35):(35)$$E_{D}\left(w\right)=\frac{1}{2}\sum_{j=1}^{n}{\left(f\left(x_{j},w\right)-\tau_{j}\right)}^{2}$$
(36)$$=\frac{1}{2}\sum_{j=1}^{n}Exp_{j}^{2}$$

Overfitting concerns and generalizing performance reduction are always present in the NN training process. Therefore, a regularization approach in the NN training process used, and a new mathematical error called the generic error is upgraded by replacing a $E_{D}\left(x\right)$ mapping error [73] depicted in Equation (Equation 37):(37)$$E\left(w\right)=\beta E_{D}\left(w\right)+\alpha E_{w}\left(w\right)$$
where
(38)$$E_{w}\left(w\right)=\frac{1}{2}\sum_{j=1}^{n}w_{j}^{2}$$

The factor $E_{w}\left(w\right)$ in Equation (Equation 38) is known as the weight decay term, and the driving parameters are α and β. They have the ability to impact the complexity and adaptability of NN. The prediction function of the ANN family is generally determined using the root mean squared error (RMSE) [74], which is defined in Equation (Equation 39):(39)$$\mathrm{RMSE}={\left(\frac{1}{n}\sum_{j=1}^{n}{\left|f\left(x_{j},w\right)-\tau_{j}\right|}^{2}\right)}^{1/2}$$

The mean absolute error (MAPE) [74] is commonly used in the electrical market, and may be defined in Equation (Equation 40):(40)$$\mathrm{MAPE}=\frac{1}{n}\frac{\sum_{j=1}^{n}\left|f\left(x_{j},w\right)-\tau_{j}\right|}{f\left(x_{j},w\right)}\times 100$$

Using the Bayesian approach, Equation (Equation 41) depicts the posterior probability distribution of w [75,76] as under:(41)$$P\left(w|D,\beta ,\alpha \right)=\frac{P(D|w,\beta )P(w|\alpha )}{P(D|\beta ,\alpha )}$$
where P(D∣w,β) is the probability function, and P(w∣α) is the prior of w. If each error item presented in Equation (Equation 42):(42)$$E_{j}=f\left(x_{j},w\right)-\tau_{j},j=1,2,\ldots ,n$$

Each error has a normal probability with zero mean and variance 1/β, the weights $w_{k},k=1,\ldots ,m$ would also be a normal with zero mean and variance 1/α [76], then
(43)$$P\left(D|w,\beta \right)=\frac{1}{Z_{D}\left(\beta \right)}\exp\left(-\beta E_{D}\left(w\right)\right)$$
(44)$$P\left(w|\alpha \right)=\frac{1}{Z_{w}\left(\alpha \right)}\exp\left(-\alpha E_{w}\right)$$

Substituting (Equation 43) and (Equation 44) into (Equation 45), the posterior becomes as:(45)$$P\left(w|D,\alpha ,\beta \right)=\frac{1}{Z_{E}\left(\alpha ,\beta \right)}\exp\left(-\beta E_{D}-\alpha E_{w}\right)$$
(46)$$=\frac{1}{Z_{E}\left(\alpha ,\beta \right)}\exp\left(-E\left(w\right)\right)$$
and the evidence function has the accompanying structure [77]:(47)$$P\left(D|\alpha ,\beta \right)=\frac{1}{Z_{D}\left(\beta \right)}\frac{1}{Z_{w}\left(\alpha \right)}\int \exp\left(-M\left(w\right)\right)dw$$
(48)$$=\frac{Z_{S}\left(\alpha ,\beta \right)}{Z_{D}\left(\beta \right)Z_{w}\left(\alpha \right)}$$

Let $w^{*}$ be the maximum value point of P(w∣D,α,β), i.e., $w^{*}$ be the minimum value point of E(w) [24]. Using the Taylor expansion of E(w) around $w^{★}$ and retaining terms up to the second order, then
(49)$$E\left(w\right)=E\left(w^{*}\right)+\frac{1}{2}\Delta w^{T}H\Delta w$$
where
(50)$$\Delta w=w-w^{*}$$

H is the Hession Matrix of E(w) at $w^{*}$. Thus, the posterior distribution can be written as in Equation(Equation 51):(51)$$P\left(w|D,\alpha ,\beta \right)=\frac{1}{Z_{E}^{*}\left(\alpha ,\beta \right)}\exp\left(-E\left(w\right)\right)$$
where $Z_{E}^{*}\left(\alpha ,\beta \right)$ is the normalization factor. While:(52)$$Z_{E}^{*}\left(\alpha ,\beta \right)={\left(2\pi \right)}^{\frac{k}{2}}{\left|H\right|}^{-\frac{1}{2}}\exp\left(-E\left(w^{*}\right)\right)$$

Picking a legitimacy premise of the weight space, with the end goal that the H is the identity I [78]:(53)$$\nabla \nabla E_{w}\left(w^{*}\right)=I$$

Setting
(54)$$\nabla \nabla E_{D}\left(w^{*}\right)=A$$
then
(55)H=βA+αI

Let $\lambda_{1},\lambda_{1},\ldots ,\lambda_{p}$ be the eigenvalues of the matrix A, then the H has eigenvalues $\lambda_{1}+\alpha ,\lambda_{2}+\alpha ,\ldots ,\lambda_{p}+\alpha$ [79], hence
(56)$$\frac{\partial }{\partial \alpha }\ln\left(|H|\right)=\frac{\partial }{\partial \alpha }\ln\left(\prod_{j=1}^{p}\lambda_{j}+\alpha \right)$$
(57)$$=\sum_{j=1}^{p}\frac{1}{\left(\lambda_{j}+\alpha \right)}$$

If the logarithm evidence for Equation (Equation 47) acquires maximization at point α, then
(58)$$\begin{array}{c} \frac{\partial }{\partial \alpha }\ln\left(P\left(D|\alpha ,\beta \right)\right)=-E_{w}\left(w^{*}\right)-\frac{1}{2}\frac{\partial }{\partial \alpha }\ln\left(|H|\right)+\\ \frac{S}{2\alpha }\frac{d\partial }{\partial \alpha }\ln\left(P\left(D|\alpha ,\beta \right)\right)=0 \end{array}$$

So,
(59)$$2\alpha E_{w}\left(w^{*}\right)=p-\sum_{j=1}^{p}\frac{\alpha }{\left(\lambda_{j}+\alpha \right)}$$
(60)$$=\sum_{j=1}^{p}\frac{\lambda_{j}}{\left(\lambda_{j}+\alpha \right)}$$
(61)=γ

The most probable values of hyper-parameters α,β are
(62)$$\alpha^{*}=\frac{1}{2}\frac{\gamma }{E_{w}\left(w^{*}\right)},\;\beta^{*}=\frac{1}{2}\frac{n-\gamma }{E_{D}\left(w^{*}\right)}$$

The predicted power consumption pattern is dispatched to the optimization module to further minimize errors and enhance accuracy.

### 3.3. Optimization Module Based on BO

The optimizing module is used to provide accurate, dependable, and robust predicting outputs by interacting with BNN to optimize hyper-parameters to generate effective and consistent predictive results. A probabilistic model is used to construct many evolutionary algorithms for this purpose. The BO algorithm has gained a lot of attention among these optimization systems.

#### BO-Based Optimizer

Significant hyper-parameters in traditional models, such as BNN, heavily depend on the datasets. Assuring a correct fit for these hyper-parameters is an art. The DL framework employs a variety of hyper-parameters’ tuning algorithms, such as grid search and random search, among others [32,33,34]. BO is a viable strategy for locating the extrema of a given objective function (OF). The OF is estimated as a Gaussian process (GP) and perceived as a proxy function (pf). BO performs well when the closed-form expression of the provided OF is unknown, but specific observations may be derived from it. In our devised model, BO is employed to find the best hyper-parameters for discovering the test or validation loss minima. The hyper-parameter search space is denoted by S, and model parameters such as the number of hidden layers are represented by $N_{h}$, dropout rate as $T_{d}$, batch size as $B_{s}$, etc. Thus, the OF can be expressed as:(63)$$F:S\left(N_{h},N_{h},N_{h},\ldots ,N_{n}\right)\subset R^{n}\to R$$

The S for determining the optimal model hyper-parameter arrangement can be characterized as $s^{*}\in S$ such that:(64)$$s^{*}=\arg\min_{s\in R}F$$

Here, the observations of the OF can be expressed as:(65)$$D_{1:m}=\left(s_{1:m},F_{s_{1:m}}\right)$$

This allows BO to develop a probabilistic model within F(s), so the model can be used to find the next position in S in a sample search, which can be found using Bayesian theory, as BO builds a posterior distribution of OF, and subsequent hyperparameter configurations are selected from this distribution. Use the initial sampling point information to calculate the shape of OF and the hyperparameters that optimize the expected results. The framework hyper-parameters in our article generate that OF, and the goal is to optimize the negative of the validation loss. BO is used to calculate critical BNN hyper-parameter values. Since these variables affect prediction accuracy and robustness, BO designed a system to select and optimize the hyper-parameters of the BNN framework. Train the BNN network with these optimum circumstances after fine-tuning it with BO (best features plus fine-tuned hyper-parameters). This is the final forecasting model that will be put to the test. The overall step-by-step procedure of the proposed framework is depicted in Figure 3.

### 3.4. BO Algorithm for Hyperparameters Tuning

As a model-based hyperparameter-tuning technique, the BO algorithm models the conditional probabilities of the validation set performance when hyperparameters are selected using surrogate functions. In contrast to the grid or random searches, the BO algorithm tracks all historical evaluations. Therefore, avoid wasting calculations to evaluate bad hyperparameters. In addition, the acquisition function finds the most promising hyperparameter to assess in the next iteration. The proposed model applies the BO algorithm strategies to find the optimal hyperparameters in the dynamic ensemble module. The BO algorithm achieves better tuning efficiency in a much shorter evaluation time. BO algorithm consists primarily of five parts: the hyperparameter space, OF, the acquisition function, the history of evaluations, and the surrogate function. In this article, we define the hyperparameters domain in Table 1. The OF is the forecasting error on the augmented validation data. We implement the tree-based Parzen window estimation (TPE) practice to accomplish the probabilistic modeling of the surrogate function and adopt the expected improvement to be the acquisition function *A*, defined in Equation (Equation 66).
(66)$$A_{g^{*}}\left(\nu \right)=\int_{-\infty }^{g^{*}}\left(g^{*}-g\right)Q\left(g|\nu \right)dg,$$
where *g* is the OF and $g^{*}$ is the threshold of OF, given the hyperparameter choice ν. The simplified algorithmic description of the TPE-based BO algorithm is shown below in Algorithm 1.
**Algorithm 1** TPE-based BOA for HP tuning.**Require:** OF*g*, TPE method M, hyperparameter domain $H_{\nu }$, acquisition function validation *A*, initialized memory U**Ensure:** OF*f*, TPE method M, HP domain $H_{\nu }$, acquisition function validation *A*, initialized memory U 1:for j=1:N do 2:Q(g∣ν)← Fit memory U using M 3:$\nu_{j+1}\leftarrow$ Maximize acquisition function *A* in Equation (Equation 66) in search for the next hyperparameter choice 4:$\;g\left(\nu_{j+1}\right)\leftarrow$ Evaluate the OF 5:$U\leftarrow U\cup \left(\nu_{j+1},g\left(\nu_{j+1}\right)\right)$ 6:End

## 4. Simulation Results and Discussion

### 4.1. Simulation Setup

The CPUs and GPUs used in this task are an Intel Core i710701 K @ 3.82 GHz and an NVIDIA GeForce RTX 2070 SUPER. Modeling, training, tuning, and testing are programmed in Python 3.7. The libraries used for this task are: statsmodels 0.12.0 (for Relief-F and Random Forest), sklearn 0.23.1 (for LR, SVR, EWTFCMSVR, BNN, and ANNMI), xgboost 1.2.1 (for XGB), Torch 1.6.0 (for ESN, MLP, LSTM, DeepAR, TCN, Nbeats, MOPSOCD, ITVPNNW, ANNMI, and AFCANN), hyperopt 0.2.3 (for BO), and pymoo 0.4.2.1 (for EDNNDEC).

### 4.2. Compared Models

The BO-algorithm-based optimization module directly correlates to between the convergence rate and accuracy. However, the devised model is better than the existing models such as ANN-MI [80], LSTM, Bi-Level [59], and AFC-ANN [81]. The above models have been determined as benchmark models due to structural resemblances with the evolved model. Convergence rate and accuracy are two quantifiable metrics to evaluate performance.

Time consumed during execution by the forecasting approach is called convergence rate, and the execution time is calculated in seconds (s).While Accuracy (A) is defined as:
(67)A=100−MAPE
and is measured in (%).

The simulation parameters are enumerated in Table 2 and are retained as the same for the presented and benchmark schemes. The explicit depiction of the simulation results is presented as follows:

### 4.3. Description of Dataset

Historical daily EL data from the publicly available PJM electricity market is used to evaluate the effectiveness of the proposed scheme. The training forecasting model is characterized by various factors (temperature, humidity, dew point, and time of day). The historic and hourly load data for the USA electrical system for the last four years (2017–2020) is used. The data includes humidity, temperature, and load parameters. The electric grid (FE) has the highest load profile and covers the most densely inhabited area. The dataset goes through the feature engineering module, where the abstracted features are extracted from the specified dataset. The subset (abstracted features) of the dataset is divided into training samples and test samples. We considered three years of data for network training and one year for network testing. The input vector, the above mentioned variables, and the main target load profile are included in the training data samples from 2017 to 2019. Test data samples are collected and used for testing in 2020. Data sample validation is created from the training sequence data to improve the parameter selection for validation errors.

### 4.4. Learning Curve Evaluation

A learning curve is a pictorial illustration that approximates the efficiency of a framework when training and testing data samples across different numbers of epochs. We can use the learning curve to notice if the selected model is training or storing data. If the bias and variance are high, the learning curve is poor, so the model does not memorize or learn. The high bias results in higher training and testing error as well as faster convergence rates. In contrast, significant variances occur when there is a substantial gap between training and test errors. In either case, the model is inappropriate, and the generalization is inadequate. Overfitting occurs when test errors begin to increase and training errors decrease. This shows that the model memorizes but does not learn. Therefore, such a model is under generalized. The dropout method and early stopping prevent overfitting problems [82]. However, for BNN, it is observed that the test errors gradually decrease, similar to the training errors in the USA power grid (FE). Therefore, the BNN model solved the problem of overfitting. In addition, the gap between training and testing errors is small, with no bias or variance, as shown in Figure 4.

### 4.5. Day-Ahead Analysis

Figure 5 shows the day-ahead ELF profile with an hourly resolution using the developed and other benchmark frameworks, such as LSTM, Bi-Level, ANN-MI, and AFC-ANN for the USA power grid (FE). The graphical representation shows that all predictive models, including the proposed model, can capture nonlinear load behavior from historical data and predict future electrical loads based on the captured behavior. It is also clear that models such as Bi-Level, MI-ANN, AFC-ANN, and LSTM use the Levenberg–Marquardt, sigmoid activation function, and multivariate AR algorithms for network training. In contrast, customized BNNs are trained with the Tangent Hyperbolic (Tanh) function due to their short execution time. This can be seen in Figure 5. The predictive load for the day ahead, based on a BNN-based model and other benchmark models, is shown in Table 3. The MAPE of the devised BNN-based framework is 0.4920%, the MAPEs of the ANN-AFC, ANN-MI, and Bi-Level models are 2.9186%, 4.3371%, and 2.4741% respectively. The developed model has lowered MAPE compared to the benchmark models, resulting in superior accuracy.

### 4.6. Convergence Rate Evaluation

Figure 6 presents a performance rating of the devised and benchmark models in relation to the rate of convergence of the USA power grid (FE). There is an inverse relation between the rate of convergence and forecast accuracy. The ANN-AFC framework is more accurate than the ANN-MI framework. This gain in accuracy reaches at the expense of more prolonged execution times. As shown in Figure 6, the execution time rises from 20 s to 110 s. The execution time of the proposed framework has been reduced for two reasons:Abstractive features are fed into the training and forecasting module, reducing network training time.The BO algorithm is used due to its significantly faster convergence rate.

Due to the adjustments made to the proposed model, the proposed STLF framework lowered the execution time from 110 s to 42 s. In contrast, ANN-MI has excellent performance in term of convergence rate, even though this model has no optimization module integrated into it. This tendency is seen well in Figure 6.

### 4.7. Scalability Analysis

Scalability research shows whether the framework under development is scalable or suitable for the scenario under consideration. Bias, threshold, input samples, and random weights are adjusted and tuned by Equation (Equation 1). These factors affect the accuracy by calculating the errors and convergence rate by calculating the execution time of the proposed framework depicted in Figure 7a,b. Forecast accuracy increases from 0 to 700 data samples and tends to stabilize while boosting samples. We can notice the effect from the value of *l* in expression (Equation 1). It is closely related to BNN training. An important value of *l* during the training process indicates fine-tuning and increases forecast accuracy. Similarly, Figure 7b shows the relationship between sample size and execution time. Using FE for feature selection, BNN for prediction, and BO algorithm for optimization, the developed model shows relatively good scalability compared to the benchmark models.

### 4.8. Computational Time Analysis

Individual models, LSTM, BNN, and Bi-Level do not integrate both FE and optimization modules, resulting in short computational times ($\tau_{c}$), and the worst error performances for daily time horizons are listed in Table 4. The performance analysis of proposed and benchmark frameworks in terms of computational time and MAPE is depicted in Figure 8a,b. However, when both the FE and the optimization modules are integrated into these individual models, the trade-off between accuracy and rate of convergence increases $\tau_{c}$ and reduces errors. The individual models, LSTM, BNN, and Bi-Level have the shortest $\tau_{c}$ of 172 s, 182 s, and 112 s, respectively, and the hybrid models (ANN-AFC, ANN-MI, and the proposed FE-BNN-BO) have $\tau_{c}$ of 510 s, 348 s, and 217 s, respectively.

When the optimization module, the FE module, or both modules are integrated with the individual predictive models, they counteract the increased $\tau_{c}$. In addition, this increase in time is due to the trade-off between convergence speed and accuracy, thereby achieving higher accuracy at the expense of excessive $\tau_{c}$. The proposed FE-BNN-BO framework reduces $\tau_{c}$ by developing changes to the BO algorithm. Therefore, the FE module modifies the functional space by removing redundant and irrelevant features, and the BO-algorithm-based optimization module adjusts the control parameters of BNN to ensure an accurate ELF.

### 4.9. Robustness Evaluation

Stochastic noises (white noise, harmonic noise, asymmetric dichotomous noise, and Lévy noise) have a great adverse effect on the prediction accuracy of electric power load. Pre-filtering real-time can effectively improve measurement accuracy. Pretreating and statistically inspecting the electric power load data is essential to characterize the stochastic noise of the electric power load. The proposed feature engineering (FE) is used to denoise load data. FE is significantly reduced stochastic noise amplitude of power load data. Therefore, the proposed time series model and FE method can effectively suppress the stochastic noise of the power load data and improve the prediction accuracy of the power load in order to maintain the robustness of the proposed model. Figure 9 shows the robustness assessment of the proposed FE-BNN-BO model and benchmark models such as LSTM, Bi-Level, ANN-AFC, and MI-ANN. The evaluation is performed by adding an error (noise) to each function and observing the accuracy of each scheme. The proposed framework is more robust than the benchmark frameworks. This is because the noise in the feature has little effect on accuracy, reducing the number of important and irrelevant features dropped during the FE phase. Therefore, the proposed FE-BNN-BO framework is also robust against functional noise.

**Remark** **1.**
*Energy consumption forecasting is of prime importance for the restructured environment of energy management in the electricity market. Accurate energy consumption forecasting is essential for efficient energy management in the smart grid (SG); however, the energy consumption pattern is non-linear with a high level of uncertainty and volatility. Keeping in view the non-linearity and complexity of the investigated problem, a BO algorithm is proposed for the optimization module of the proposed model to further improve accuracy with reasonable convergence of the forecasting results returned from the BNN-based forecaster. The proposed FE-BO-BNN model is examined on FE power grid data from the USA in terms of MAPE and convergence rate. Simulation results validated that the proposed FE-BO-BNN model achieved 0.4920 accuracy in terms of MAPE, which is better than the benchmark models, such as Bi-Level (2.4721), AFC-ANN (2.9286), and MI-ANN (4.3371). The proposed model reduced the average execution time by 21.1%, 35.5%, and 61% when compared to MI-ANN, AFC-ANN, and Bi-Level, respectively. It is concluded that our proposed FE-BO-BNN model outperformed benchmark electrical-energy-consumption forecasting models in terms of both accuracy and convergence rate.*


## 5. Conclusions

ELF is an essential component of the reliable operation of the energy system, since accurate LF is helpful in reducing the generation-demand mismatch through optimal decision-making and advance planning. However, the short- and/or long-term power generation or infrastructure planning depends on accurate forecast results with the possibility of marginal error, although a great effort is being given to the development of accurate forecasting algorithms. However, there is still a possibility to further improve the algorithmic accuracy by considering their control parameters, since the performance and accuracy depend on these control parameters. In this regard, the paper has presented a new and hybrid load-forecasting model based on BNN and BO. The proposed framework has used the BO algorithm to fine-tune the hyper-parameters of BNN to improve its accuracy. The FE module is integrated into the BNN model to further improve the computational efficiency and solve the problem of model dimensionality reduction. Through this combination, the proposed model simultaneously achieves higher stability, convergence, and accuracy. The devised framework is assessed using an hourly load dataset obtained from the USA energy grid (FE). The devised model is evaluated and compared with other latest models such as Bi-Level, ANN-AFC, ANN-MI, and LSTM, considering accuracy and convergence rate. In other words, the proposed ELF model outperforms Bi-Level by 15.73%, MI-ANN by 29.1%, and AFC-ANN by 3.97%.

## Figures and Tables

**Figure 1 sensors-22-04446-f001:**
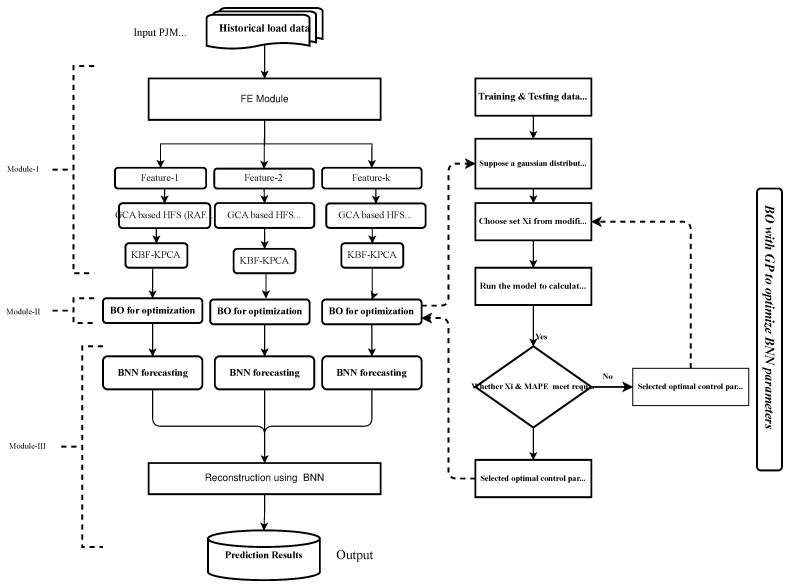
Single-line diagram of a new hybrid proposed model FE-BO-BNN for ELF.

**Figure 2 sensors-22-04446-f002:**
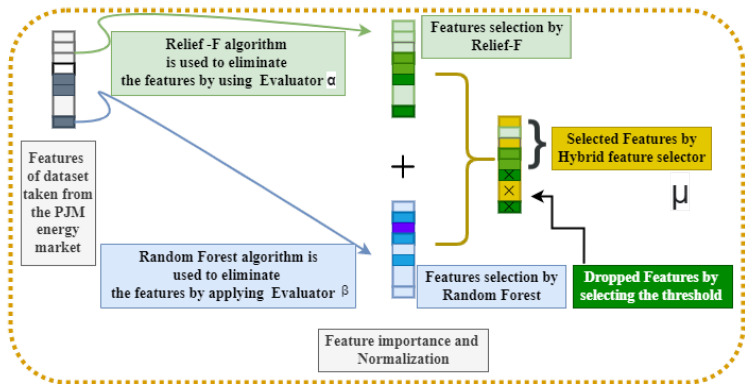
Illustration the details of fused feature selector (FFS) if a feature is reserved by an index, which is given by Relief-F and Random Forest.

**Figure 3 sensors-22-04446-f003:**
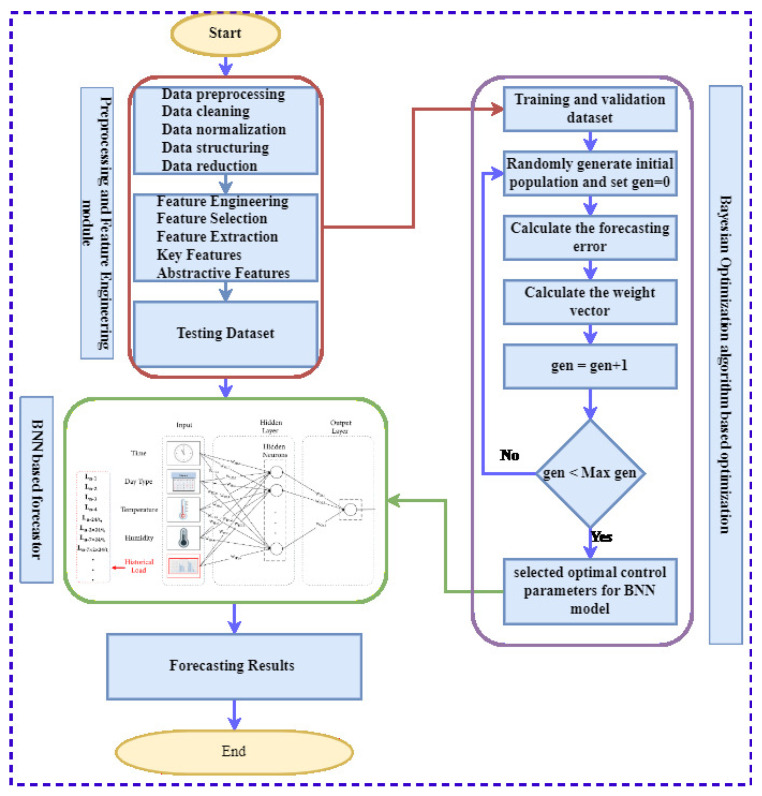
Step-by-step working flow chart of the proposed schematic framework for ELF. Red box shows data pre-processing and feature engineering module, green box shows BNN-based forecaster module, and purple box represents BO-algorithm-based optimization module.

**Figure 4 sensors-22-04446-f004:**
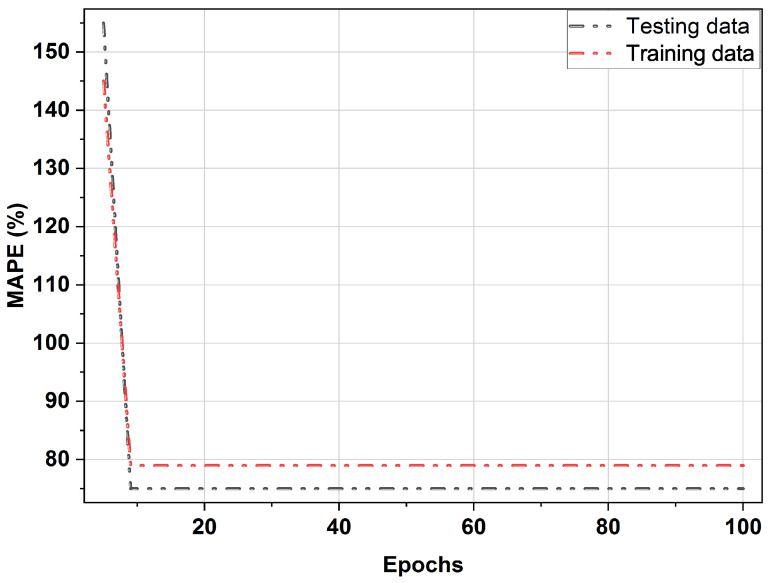
The BNN-EMC learning curve evaluation on FE.

**Figure 5 sensors-22-04446-f005:**
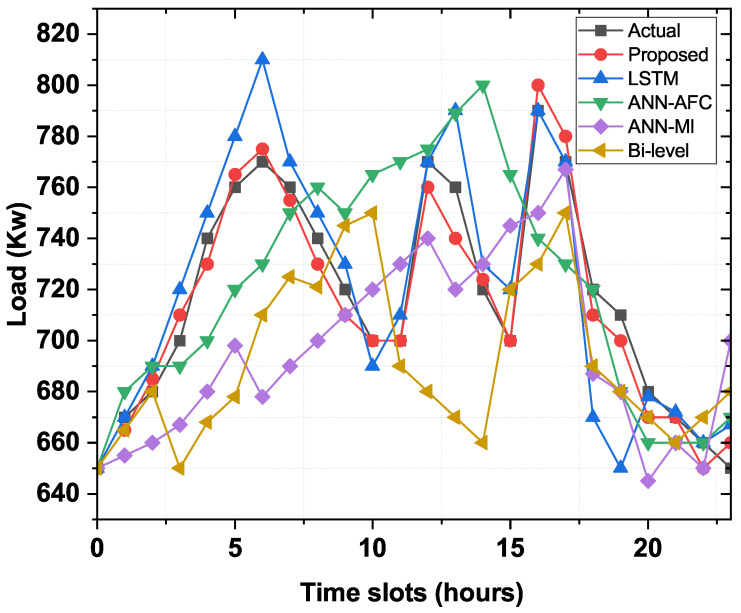
Actual and predicted load day-ahead assessment in terms of prediction accuracy.

**Figure 6 sensors-22-04446-f006:**
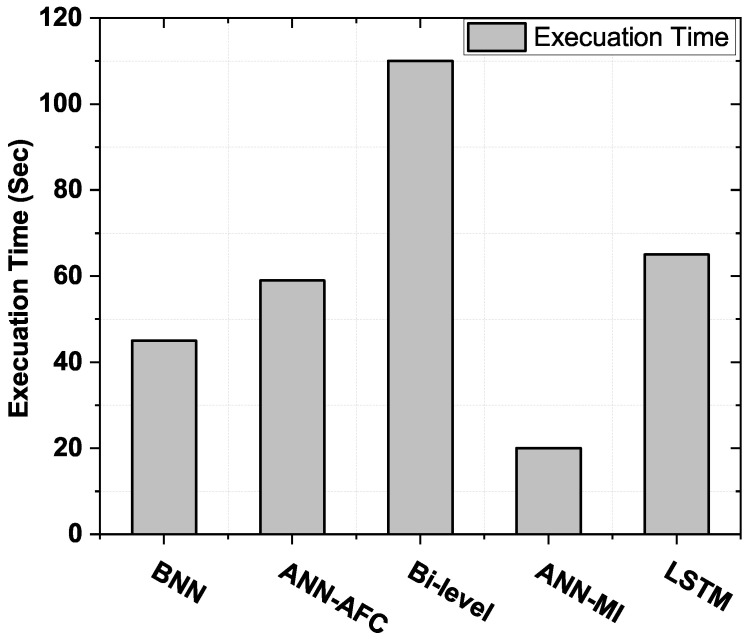
Convergence rate of daily load data of USA power grid (FE).

**Figure 7 sensors-22-04446-f007:**
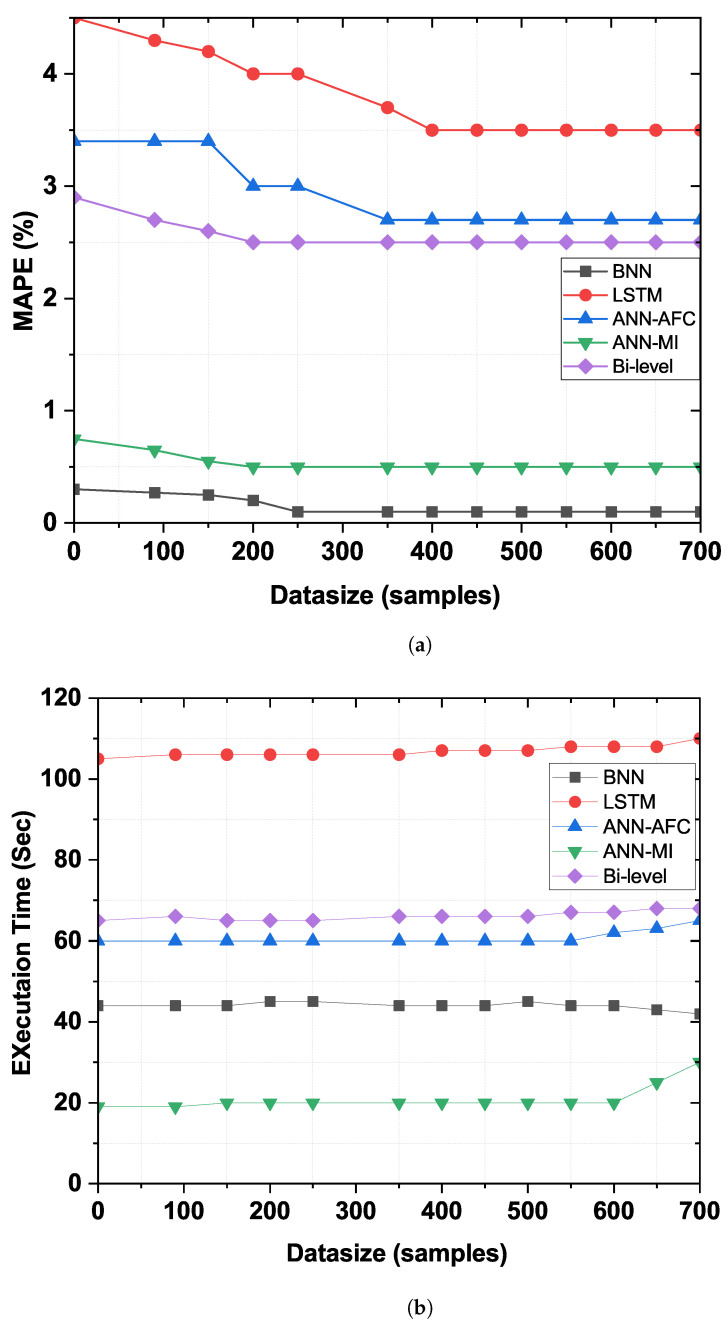
Scalability evaluation of the proposed model and benchmark models in terms of (**a**) accuracy and (**b**) convergence rate.

**Figure 8 sensors-22-04446-f008:**
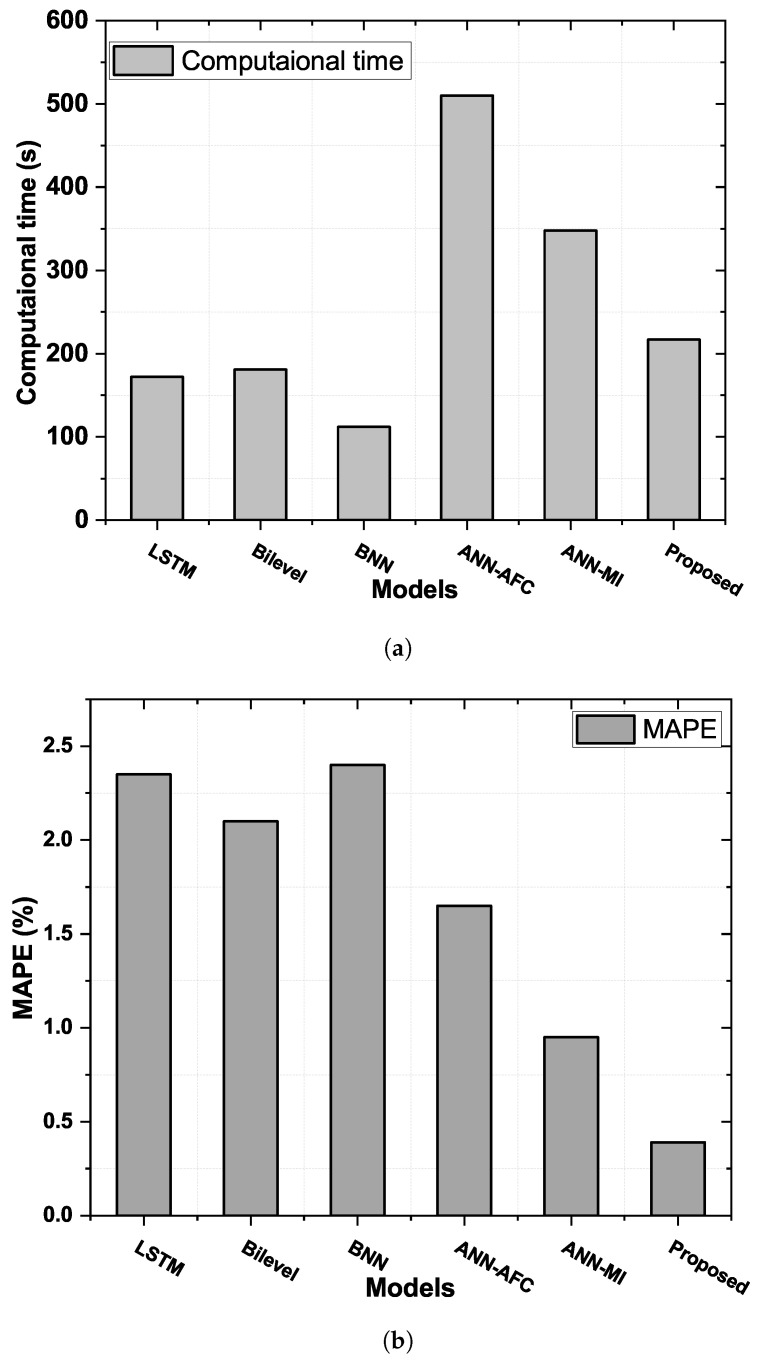
Performance analysis of proposed and benchmark frameworks in terms of computational time and MAPE. (**a**) Scalability evaluation of the devised and benchmark frameworks by error performance. (**b**) Comparative analysis of models with and without FE and optimization algorithm in terms of MAPE (%).

**Figure 9 sensors-22-04446-f009:**
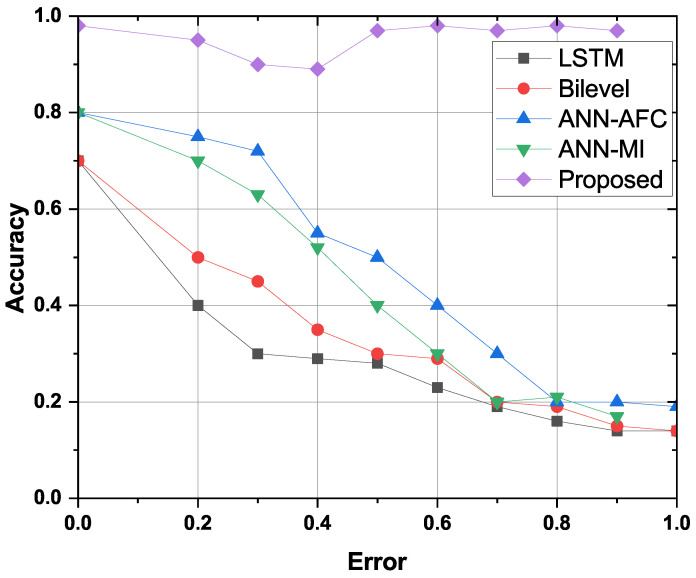
Robustness analysis of devised and other benchmark models.

**Table 1 sensors-22-04446-t001:** Hyperparameter domain of the BO-algorithm-based tuning for the dynamic ensemble configurations.

Hyperparameters	Notation	Hyperparameter Space
Cold-start model index	$\iota_{cs}$	η(1,10)
Maximum number of chosen models	$N_{max}$	η(1,12)
Size of past observations to evaluate	χ	η(1,10)
Weight-calculation function	ω	Choice of $\left(\frac{{\left|\cdot \right|}^{-1}}{{\sum |\cdot |}^{-1}},\frac{{\left|\cdot \right|}^{-2}}{{\sum |\cdot |}^{-2}},\frac{\exp(-|\cdot |)}{\sum \exp(-|\cdot |)}\right)$
Discount factor of past observations	κ	ν(1,1.5)
Updated parameter	ζ	ν(0,1)

η denotes discrete uniform distribution; ν denotes uniform distribution.

**Table 2 sensors-22-04446-t002:** Simulation parameters.

Control Parameters	Value
Hidden layer	3
Neurons in hidden	15
Output layer	1
Number of output neurons	1
Number of epochs	200
Number of iterations	200
Learning rate	0.0017
Momentum	0.55
Initial weight	0.1
Initial bias	0
Max	0.8
Min	0.2
Decision variables	3
Population size	23
Delay of weight	0.0003
Historical load data	4 yrs
Exogenous parameters	4 yrs

**Table 3 sensors-22-04446-t003:** Targeted and forecasted load (F.load) of the devised and benchmark frameworks evaluated by using MAPE.

Proposed and Benchmark ELF Frameworks									
**Hours**	**Targeted (kW)**	**Proposed**	**Bi-Level**		**ANN-AFC**		**ANN-MI**		
		**F.load**	**MAPE**	**F.load**	**MAPE**	**F.load**	**MAPE**	**F.load**	**MAPE**
		(**kW)**	(**%)**	(**kW)**	(**%)**	(**kW)**	(**%)**	(**kW)**	**(%)**
00.00	670.3223	667.1829	0.4525	673.4829	1.5400	645.5829	3.9157	650.8923	3.1255
01.00	677.7923	674.4538	0.4926	660.4538	2.5581	665.4538	1.8204	630.7923	6.9343
02.00	700.3192	703.7687	0.4926	715.7687	4.6806	635.7687	9.2173	725.3192	3.5698
03.00	734.5654	738.1835	0.4926	725.1835	2.0696	745.1835	1.4455	759.5654	3.4034
04.00	760.9115	757.1637	0.4924	743.1637	1.5923	777.1637	2.1359	740.9115	2.6284
05.00	767.4346	771.2146	0.4925	755.2146	4.4182	795.2146	3.6199	797.4346	3.9091
06.00	754.7077	758.4250	0.4915	730.4250	4.0749	745.4250	0.8851	750.7077	2.3554
07.00	744.3962	748.0627	0.4923	714.0627	4.3205	755.0627	1.2300	740.3962	0.5300
08.00	731.4692	727.8664	0.4925	699.8664	1.7461	718.8664	1.4329	735.4692	0.5373
09.00	717.9577	714.4214	0.4926	705.4214	0.7822	700.4214	1.7229	720.9577	0.5468
10.00	706.0231	709.5006	0.4926	700.5006	1.4930	730.5006	2.4425	760.0231	6.4179
11.00	699.6500	703.0961	0.4925	710.0961	3.9058	699.0961	0.0792	685.6500	2.0010
12.00	703.1462	706.6095	0.4925	730.6095	2.8131	725.6095	3.1947	707.6213	0.9955
13.00	726.0346	729.6107	0.4926	705.6107	2.2272	750.6107	3.3850	710.1462	1.9283
14.00	753.6077	757.3196	0.4925	755.3196	1.1835	700.3196	7.0711	740.0346	1.5923
15.00	768.8000	772.5867	0.4925	785.5867	3.6695	785.5867	2.1835	765.6077	1.2435
16.00	768.8538	772.6408	0.4925	740.6408	4.5999	778.6408	1.9500	720.8000	6.6503
17.00	754.7423	751.0248	0.4925	720.0248	1.1768	740.0248	0.1325	763.8538	2.1325
18.00	730.7462	734.3454	0.4926	739.3454	1.5246	690.9239	1.3136	755.7423	3.7790
19.00	703.3885	699.9239	0.4926	715.9239	2.1544	670.9967	1.7721	743.7462	8.8145
20.00	682.3577	678.9967	0.4925	760.9967	1.2358	655.1795	1.6650	765.3885	0.1153
21.00	661.9192	665.1795	0.4926	676.1795	2.1540	630.0057	1.0182	682.3577	2.1151
22.00	672.6923	676.0057	0.4926	681.0057	1.2822	630.1417	6.3456	675.9192	0.4797
23.00	676.6923	680.1750	0.4926	686.5057	1.4502	633.1530	5.8778	680.6923	1.1893
Avg.			0.4920		2.4741		2.9186		4.3371

**Table 4 sensors-22-04446-t004:** Analysis of BNN model with and without FE and optimization modules for ELF.

Models without FE and Optimization Modules						Models with FE and Optimization Modules					
**LSTM**		**Bi-Level**		**BNN**		**ANN-AFC**		**ANN-MI**		**Proposed**	
$\tau_{c}$ **(s)**	**MAPE** **(%)**	$\tau_{c}$ **(s)**	**MAPE** **(%)**	$\tau_{c}$ **(s)**	**MAPE** **(%)**	$\tau_{c}$ **(s)**	**MAPE** **(%)**	$\tau_{c}$ **(s)**	**MAPE** **(%)**	$\tau_{c}$ **(s)**	**MAPE** **(%)**
172	2.31	182	2.1	112	2.3	510	1.45	348	0.95	217	0.39
240	2.31	276	2.1	187	2.3	579	1.45	408	0.95	265	0.39
308	2.31	321	2.1	245	2.3	456	1.45	456	0.95	412	0.39

## Data Availability

Not applicable.

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
