# Peer review of "Hyperparameter Optimization of Bayesian Neural Network Using Bayesian Optimization and Intelligent Feature Engineering for Load Forecasting"

_sensors, 2022, doi:10.3390/s22124446_

Round 1

Reviewer 1 Report

The topic is interesting and this paper is ambitious with a strong desire to consider important aspects in the field of electric load forecasting. This paper proposes a hybrid framework through combining the Feature Engineering (FE) and Bayesian Optimization (BO) algorithms with BNN. Through this combination, the proposed model simultaneously achieves the higher stability, convergence, and accuracy. The content of the manuscript is good. However, the paper requires revisions before it can be accepted.

  1. Please, rewrite the context of the abstract. Abstract should concisely state what was done, how it was done, principal results, and their significance. A single paragraph of about 200 words maximum.
  2. Authors should cover the latest findings in area where the contribution was done (a recent literature review is needed), so that we can see what is the improvement compared to existing recent works.
  3. The readability of the manuscript is little bit poor. Numerous grammar and syntax errors along with typos are present in this version of the paper. Authors are advised to thoroughly revise the whole manuscript. The quality of all figures should be improved.
  4. How does this framework deal with loads under consumption uncertainties?
  5. Is the power consumption measured just in terms of real power? Energy consumption patterns must be in terms of active and reactive power.
  6. The test case is not described in enough details. Please highlight the single line diagram and technical specifications of the different elements.
  7. It is suggested to add flow chart of the proposed framework, and the software used in the implementation of the different components in test system should be mentioned.

Author Response

Original Manuscript ID: Sensors-1736044

Original Article Title: "HYPERPARAMETER OPTIMIZATION OF BAYESIAN NEURAL NETWORK USING BAYESIAN OPTIMIZATION AND INTELLIGENT FEATURE ENGINEERING FOR LOAD FORECASTING."

To: Editor Sensor

Re: Response to reviewers

Dear Editor,

Thank you for allowing a resubmission of our manuscript, with an opportunity to address the reviewers' comments.

We are uploading (a) our point-by-point response to the comments (below) (response to reviewers), (b) an updated manuscript with highlighting indicating changes, and (c) a clean, updated manuscript without highlights (PDF main document).

Best regards,

Muhammad Babar Rasheed et al.

Note: -

  1. Responses of Reviewer # 1 and common to Reviewer # 2 are given blue color.
  2. Responses of Reviewer # 2 are given red color.

Reviewer#1, Concern #1: Please, rewrite the context of the abstract. The abstract should concisely state what was done, how it was done, the principal results, and their significance. A single paragraph of about 200 words maximum.

Author response:  Thank you for your kind concern. The abstract has been rewritten in the revised version.

This paper proposes a new hybrid framework for short-term load forecasting (STLF) by combining the Feature Engineering (FE) and Bayesian Optimization (BO) algorithms with Bayesian Neural Network (BNN). The FE module comprises feature selection and extraction phases. Firstly, by merging the Random Forest (RaF) and Relief-F (ReF) algorithms, we developed a hybrid feature selector based on grey correlation analysis (GCA) to eliminate the feature redundancy. Secondly, the feature extraction module integrates a radial basis Kernel function and principal component analysis (KPCA). Thirdly, the Bayesian optimization (BO) algorithm is used to fine-tune the control parameters of the BNN and provides more accurate results by avoiding the optimal local trapping. The proposed FE-BNN-BO framework works in such a way to ensure stability, convergence, and accuracy. The proposed FE-BNN-BO model is tested on hourly load data obtained from the PJM, USA electricity market. In addition, the simulation results are also compared with the other benchmark models such as bi-level, long short-term memory (LSTM), accurate and fast convergence-based ANN (ANN-AFC), and mutual information-based ANN (ANN-MI). The results show that the proposed model has significantly improved the accuracy with a fast convergence rate and reduced the mean absolute percent error (MAPE).

Author action: We updated the manuscript considering your kind concern. The changes can be seen in the revised version with blue highlights [Page# 1].

Reviewer#1, Concern # 2: Authors should cover the latest findings in an area where the contribution was done (a recent literature review is needed) so that we can see what the improvement is compared to existing recent works.

Author response: Dear Reviewer, thank you very much for pointing this out. We have included the recent and relevant literature to cover the latest findings to identify the novelty of our work. We updated the introduction and literature survey sections supporting the relevant and current references.

Author action: We updated the manuscript considering your comments. The revised version can see the changes [Pages # 1, 2, 3, 4, 5, and 6].

Reviewer#1, Concern # 3: The manuscript's readability is a little bit poor. Numerous grammar and syntax errors, along with typos, are present in this version of the paper. Authors are advised to thoroughly revise the whole manuscript. The quality of all figures should be improved.

Author response: Dear Reviewer, we have thoroughly revised the manuscript to make it more understandable for the reviewers and other interested readers. Furthermore, to improve the quality, we modified the contribution, methodology, proposed model, results, discussion, concluding remarks, and conclusion sections with other results in graphical and tabular forms.

Author action: We updated the manuscript considering your comments.

Reviewer#1, Concern # 4: How does this framework deal with loads under consumption uncertainties?

Author response:  The sophistication of load performance tools has significantly increased the number of user inputs and parameters used to define electricity load models. There are numerous sources of consumption uncertainty in model parameters that exhibit varied characteristics. Therefore, the consumption uncertainty analysis is crucial to ensure the validity of the proposed model through simulation results when analysing and predicting the performance of complex energy systems, especially in the absence of adequate experimental or real-world data. Furthermore, different types of uncertainties are often propagated using similar methods, which leads to a false sense of validity. A comprehensive framework to systematically identify, quantify and propagate these uncertainties is always sought. To handle this uncertainty, this paper used a Bayesian optimization (BO) algorithm to efficiently solve a class of robust optimization problems through control parameter tuning that may arise during load forecasting. The central idea is to use the Gaussian process models of loss functions (or robustness metrics) and appropriate acquisition functions to guide the search for a robust optimal solution. Where the proposed framework connects the fundamentals of uncertainty quantification and energy analysis features. As a result, the proposed model (FE-BNN-BO) is designed to improve the forecasting accuracy with the opportunity to handle uncertain parameters through the BO module.

Author action: We updated the manuscript considering your comments.

Reviewer#1, Concern # 5: Is the power consumption measured just in terms of real power? Energy consumption patterns must be in terms of active and reactive power.

Author response: Dear Reviewer, for the implementation and validation of the proposed model, this work has used the “hourly load data obtained from the PJM, USA electricity market.” The dataset comprises only the real power instead of both real and reactive powers. That is why the obtained results reflect the load variations based on real power. The main reason for the selection of such type of a dataset is given below:

Power load forecasting, in which the aim is to predict future power loads based on historical power loads datasets (active power load and reactive power load), has become a topic of great interest in both research and engineering fields in the past two decades.  Since the power factor constrains the relationship between the active and reactive loads, the reactive load can be estimated from the active load. However, the power factor is not a constant value but varies over time. Thus, it may be difficult to accurately deduce the reactive load from the active load and vice versa. Another natural idea is to directly extend the methods designed for active load forecasting to reactive load forecasting; however, it is difficult to achieve ideal results because of the different characteristics of the two kinds of loads. Since the active load is easily affected by external factors such as weather due to its relationship with the user operation situation. Accordingly, the methodologies applied for active load forecasting mostly rely on analysing the past loads in combination with multiple contemporaneous external factors to predict the future loads through statistical analysis or artificial intelligence [9].

In contrast, the response of reactive power to external factors is not as sensitive as that of active power. In addition, reactive power is local and decentralized, and its power flow direction is not as clear as active power.  This research considers hourly active power load data obtained from the USA energy grid taken from the publicly available PJM electricity market.

Author action: We updated the version with blue highlights. The changes can be seen in the revised version.

Reviewer#1, Concern # 6: The test case is not described in enough detail. Please highlight the single-line diagram and technical specifications of the different elements.

Author response:  The test case has been presented and discussed in more detail in the revised version.

The modifications can be seen in section # 4, sub-sections 4.3, 4.4, 4.5, 4.6, 4.7, and 4.8, and Figs. 7 and 8. At the same time, the single-line diagram is included in the revised version (Fig. 1).

Fig. 1: Single line diagram of the proposed framework FE-BO-BNN.

Author action: We updated the manuscript considering your comments. The revised version can see the changes [Pages # 6, 17, 18, and 19].

Reviewer#1, Concern # 7: It is suggested to add a flow chart of the proposed framework, and the software used in the implementation of the different components in the test system should be mentioned.

Author response:

  1. We have added the flow chart of the proposed framework in Fig. 3. [Page# 13].
  1. Modeling, training, tuning, and testing are programmed in Python 3.7. The libraries used for this task are: statsmodels 0.12.0 (for ReliefF and Random Forest), sklearn 0.23.1 (for LR, SVR, EWTFCMSVR, BNN, ANNMI), xgboost 1.2.1 (XGB) For), Torch 1.6.0 (for ESN, MLP, LSTM, DeepAR, TCN, Nbeats, MOPSOCD, ITVPNNW, ANNMI, AFCANN), hyperopt 0.2.3 (BO) and pymoo 0.4.2.1 (EDNNDEC).

Author action: We updated the manuscript considering your comments. The revised version can see the changes [Pages # 13,14, and 15].

Reviewer 2 Report

The paper presents a hybrid framework combining Feature Engineering (FE) and Bayesian Optimization (BO) with Bayesian Neural Network (BNN) to characterize nonlinear time-series predictions. The performance of the proposed FE-BNN-BO model is validated by comparing the results with the existing models in terms of mean absolute percentage error (MAPE). The results confirm the proposed methodology. 

In general, the article should be handled carefully from the beginning (including revision of English style/grammar and typos). The results should be compared with the same research study in the literature. And the article organization format should be revised. Here are some additional questions/points to be addressed in the paper:

1. The authors have included a session of “Literature Survey”; however, the authors should consider extending the discussion on related works in terms of results not only describing the models.  

2. A lot of typos throughout the article, for example
“In contrast, improve the accuracy of the algorithm, is it also required to better optimize and tune the control variables.”
“This study proposes a navel hybrid framework based on the FE method, neural network model (BNN)”
“Therefore, the original sequence satisfies the “larger is better" characteristic. [55], The actual data sequence is normalized as follows:”
“and ν a distinguishing coefficient”
“Scalability evaluation of The devised and benchmark frameworks by error performance”

3. Although Figures 1 and 2 are illustrative, they are not discussed in the corpus of the article. Besides, Figure 1 should have more details about the algorithms. What is Rf and RF in Figure 1 (should be RaF and ReF?)

4. The authors should use the pseudo-code with the MDPI’s format.

5. The paragraph “The dataset is divided into training samples and test samples (extracted features). Three years of data will be used for network training and one year for network testing. The input vector, the above variables, and the main target load profile are included in the training data sample from 2017 to 2020. Test data samples were collected and used for testing in 2020.” presents two different information. The first sentence indicates that the train dataset was composed of 3 years (i.e., 2017, 2018, and 2019). The question is, was the model trained with all the dataset, or only with 3 years? 

6. The authors did not mention the computational requirements for all the models. Did the authors tune the parameters and optimized the settings during training the weights (e.g., number of epochs, learning rate, optimizer, batch size, drop rate, etc.)?

7. What was the library and language used? Python (Tensorflow, Keras, PyTorch)? MATLAB?

8. The authors should suggest an explanation of why the proposed model maintains high accuracy in the “Robustness Evaluation” when compared with the benchmark models. Besides, what kind of error was aggregated to the time series (additive or multiplicative noise?) (uniform noise or Gaussian noise?)

9. The authors point that “The performance of the proposed FE-BNN-BO model is validated by comparing the results with the existing models in terms of mean absolute percentage error (MAPE), variance, and correlation”. I couldn't find the “variance, and correlation” analysis in the paper.

Round 2

Reviewer 1 Report

The reviewer has no further comments; authors have satisfactorily addressed most of my comments.

Reviewer 2 Report

The paper is suitable for Sensors.